# Hydatid Disease: A Radiological Pictorial Review of a Great Neoplasms Mimicker

**DOI:** 10.3390/diagnostics13061127

**Published:** 2023-03-16

**Authors:** Sultan Abdulwadoud Alshoabi, Abdulaziz H. Alkalady, Khaled M. Almas, Abdullatif O. Magram, Ali K. Algaberi, Amal A. Alareqi, Abdullgabbar M. Hamid, Fahad H. Alhazmi, Abdulaziz A. Qurashi, Osamah M. Abdulaal, Khalid M. Aloufi, Walaa M. Alsharif, Kamal D. Alsultan, Awatif M. Omer, Awadia Gareeballah

**Affiliations:** 1Department of Diagnostic Radiology Technology, College of Applied Medical Sciences, Taibah University, Almadinah Almunawwarah 42353, Saudi Arabia; 2Advanced AlRazi Diagnostic Center, Al-Hodeidah 86XC+C75, Yemen; 3Berlin Scan Center, Ibb, Yemen; 4Typical Doctors Center, Taiz, Yemen; 5Radiology Department, 21 September University of Medical and Applied Science, Sana’a, Yemen; 6Radiology Department, Rush University Medical Center, Chicago, IL 60612, USA

**Keywords:** larval stage of *Echinococcus granulosus*, unilocular simple cyst, cyst with floating membranes, cyst with daughter cysts, spoke-wheel appearance, cyst with waterlily sign, calcified cyst

## Abstract

Hydatid cyst is a common name for the larval stage of a tapeworm species of *Echinococcus granulosus*, which is transmitted from animals to humans via the fecal–oral route. Hydatid cysts predominantly affect the liver (75%), followed by the lung (15%), and they can affect many organs in the human body. Medical imaging modalities are the keystone for the diagnosis of hydatid cysts with high sensitivity and specificity. Ultrasound imaging with high resolution is the first choice for diagnosis, differential diagnosis, staging, establishing a role in interventional management, and follow-up, and it can differentiate Type I hydatid cysts from simple liver cysts. Unenhanced computed tomography (CT) is indicated where or when an ultrasound is unsatisfactory, such as with chest or brain hydatid cysts, when detecting calcification, and in obese patients. Magnetic resonance imaging (MRI) is superior for demonstrating cyst wall defects, biliary communication, neural involvement, and differentiating hydatid cysts from simple cysts using diffusion-weighted imaging (DWI) sequences. According to the phase of growth, hydatid cysts occur in different sizes and shapes, which may mimic benign or malignant neoplasms and may create diagnostic challenges in some cases. Hydatid cysts can mimic simple cysts, choledochal cysts, Caroli’s disease, or mesenchymal hamartomas of the liver. They can mimic lung cystic lesions, mycetoma, blood clots, Rasmussen aneurysms, and even lung carcinomas. Differential diagnosis can be difficult for arachnoid cysts, porencephalic cysts, pyogenic abscesses, and even cystic tumors of the brain, and can create diagnostic dilemmas in the musculoskeletal system.

## 1. Introduction

Echinococcosis is a zoonotic parasitic infection caused by the larval stage of the tapeworm (cestode) species of the genus Echinococcus, which is transmitted to humans from animals [1]. The two forms of medical importance are cystic echinococcosis (CE), which is caused by the larval stage of *Echinococcus granulosus* (*E. granulosus*), and alveolar echinococcosis (AE), which is caused by the larval stage of *Echinoccus multilocularis* (*E. multilocularis*) and has a high enough mortality rate to be called “worm cancer” [1,2]. The World Health Organization (WHO) estimates the incidence of human infection to be more than 50/100,000 persons annually in echinococcosis endemic areas. The prevalence is high in parts of Argentina, Central Asia, and East Africa [3].

Echinococcosis is transmitted by the fecal–oral route through either direct contact with infected definitive animal hosts, such as dogs, or through the ingestion of parasitic eggs in contaminated food, water, or soil [4]. CE is the most frequent type, and the adult tapeworm of *E. granulosus* lives in the intestine of the definitive host. The larval phase of *E. granulosus* develops in intermediate hosts, including humans, after infection by the ingested eggs of the parasite. The hydatid cyst, or hydatidosis, is a common name for the larval phase of *E. granulosus* [5].

The ingested embryonated eggs of the parasite reach the intestine and enter the portal vein to the liver, which acts as the first filter and stops about 75% of the embryos, followed by the lungs [6]. Figure 1.

Hydatid disease is highly endemic in many livestock-rising areas worldwide, and it is still neglected in scientific research. In the literature, we found a few previous pictorial reviews with old dates, and most of the other publications regarding this topic were case study reports. The current pictorial review aims to revise the common medical imaging features of hydatid cysts appearing on different imaging modalities in different human organs. In this review, we used the available images for hydatid cysts from our work in three centers in Yemen, which is considered an endemic area with this parasite. Due to the paucity of pictorial reviews regarding hydatid cysts, this review will be beneficial for physicians, radiologists, and medical imaging residents who are interested in the diagnosis of hydatid cysts. Figure 2.

## 2. Diagnosis of Hydatid Cyst

Hydatid disease is diagnosed using medical imaging modalities, including abdominal ultrasound imaging, an X-ray of the chest, and computed tomography (CT) of the abdomen, chest, and brain. Serological antibody-detecting essays using diverse native antigens are only used to confirm the diagnosis because of the great difference in their sensitivities and specificities. An enzyme-linked immunosorbent assay (ELISA) using the synthetic peptide p176 has demonstrated a good performance in diagnosing hydatid disease, with up to 80% and 93% sensitivity and specificity, respectively [7]. Ultrasound imaging is the first choice due to its availability, lack of radiation, high resolution for diagnosis, differential diagnosis, staging, establishment of a role in interventional management, follow-up, and screening to assess the prevalence of abdominal hydatid cysts. Unenhanced CT is indicated where or when an ultrasound is unsatisfactory, such as in chest or brain hydatid cysts, when detecting calcification, and in obese patients [8]. Figure 3.

## 3. Structure of the Hydatid Cyst

Hydatid cysts have the following three layers: (1) The outer layer (pericyst) consists of modified host cells, including fibroblasts, giant cells, and eosinophils, forming a fibrous and protective zone. (2) The middle laminated acellular membrane, which allows passage of the nutrient. (3) The thin inner germinal layer. The middle laminated layer and the inner germinal layer form the true wall of the hydatid cyst, called the endocyst. The thickness of these layers tends to be the thickest in the liver compared to other organs. The acellular laminated layer is occasionally called an ectocyst [9]. The pericyst is also known as the ectocyst or adventitial layer [10]. The infectious embryonic tapeworm “scolices” develop from an outpouching of the germinal layer [9].

## 4. Classification of Hydatid Cysts

I: The WHO Informal Working Group on Echinococcosis (WHO-IWGE) classification of hydatid cysts assigned six cyst stages into three clinical groups as follows: (1) The “active” group of developing cysts, which may be unilocular (CE1), or multivesicular with daughter vesicles (CE2), which are usually fertile cysts containing viable protoscoleces. (2) The “transitional” group, which may be cysts with a detachment of the endocyst membrane (CE3a), or predominantly solid cysts with daughter vesicles inside it (CE3b). (3) The “inactive” group includes solid contents (CE4), or solid contents with calcification (CE5), which are almost always nonviable. The WHO classification provides a rational basis for choosing the appropriate treatment scheme and follow-up [8,11]. Figure 4, Figure 5 and Figure 6.

II: Based on the medical imaging morphology, hydatid cysts are classified into four types, as follows: [9,12,13].

Type I: Simple cyst with no architecture: This type constitutes the initial and active phase of hydatid disease in which the three layers are intact [9]. During the initial stage of development of the hydatid cyst, it appears as a well-circumscribed, unilocular cystic lesion with no internal architecture, with or without internal septations or hydatid sand, with frequent enhancement of the cyst wall and septa on post-contrast CT [12]. On magnetic resonance imaging (MRI), hydatid cysts appear as a low signal intensity on T1-weighted images (T1WIs), and a high signal intensity on T2-weighted images (T2WIs) with a low signal intensity rim “rim sign”. The DWI sequence can differentiate Type I hydatid cysts from simple liver cysts [13]. Figure 7.

Type II: Cyst with daughter cysts and matrix: This type represents the active phase in the parasite life cycle and in the dissemination of hydatid disease [9]. Cysts with multiple septa representing the walls of the daughter cysts inside the mother cyst are usually arranged in the periphery. According to the maturity and arrangement of the daughter cyst, Type II can appear as follows: (1) Type IIA contains multiple daughter cysts arranged at the periphery of the mother cyst with a high density central matrix forming a “wheel spoke appearance”, (2) Type IIB contains multiple irregular daughter cysts occupying the cyst forming “rosette appearance”, and (3) Type IIC contains a hyperdense matrix and occasionally calcification or daughter cysts [12]. On an MRI, daughter cysts appear on a low signal intensity or isointense relative to the matrix on both T1WIs and T2WIs [13]. Figure 8, Figure 9, Figure 10 and Figure 11.

Type III: Calcified cysts: This phase constitutes the inactive dead phase of hydatid disease, which cannot spread, has no mass effect or complications, and does not require surgery [9]. It appears as a calcified lesion with posterior acoustic shadowing on ultrasound imaging, a round hyperdense lesion on CT, and a low signal intensity area on an MRI [13]. CT is the gold standard imaging modality used to diagnose calcified hydatid cysts [12]. Figure 12 and Figure 13.

Type IV: Complicated hydatid cyst: This is a hydatid cyst with a rupture or superinfection, which may be seen in both Types I and II [13]. CTs and MRIs play a major role in diagnosing complicated hydatid cysts [12,13]. The rupture of hydatid cysts occurs in 50% of cases, mainly due to age degeneration of the parasitic membrane or a defense mechanism [13]. Complications of hydatid cysts include the following: (1) Mass effect: This is when a cyst reaches a large size, which can cause biliary duct dilatation either by compression of the nearby duct or by perforation into the biliary duct. (2) Rupture of hydatid cyst: A rupture may be internal, communicating with the passage of the cystic contents into the biliary ducts, or direct, when cystic content spillage into the peritoneal cavity causes disseminated disease. (3) Hydatid disease infection is generally seen in ruptured hydatid cysts, which permits bacteria to pass easily into the cyst. Air within the cyst cavity is a clue, as it is thick and enhances walls after contrast administration on CTs and MRIs. (4) Exophytic growth occurs via the bare area of the liver with transdiaphragmatic migration to the lung or mediastinum, or via the gastrohepatic ligament into the peritoneal cavity. (5) Peritoneal seeding is almost always secondary to hepatic hydatid disease. It occurs due to previous hepatic hydatid surgery or after a spontaneous or traumatic rupture. CTs and MRIs are valuable imaging modalities for diagnosing peritoneal hydatid cysts [9]. Figure 14, Figure 15, Figure 16, Figure 17 and Figure 18.

## 5. Discussion

Hydatid cysts predominantly affect the liver (75%) [6,14], followed by the lung (15%) [6], and they are less common in the spleen, kidneys, and brain [15].

### 5.1. Hydatid Cysts of the Liver

The liver is the most affected organ in the body because it is the first filter of portal venous blood and it stops about 75% of ingested embryonated eggs [6]. Hydatid cysts of the liver are often asymptomatic and often represent an incidental finding on medical imaging; however, manifestations may occur due to cyst expansion, leading to hepatomegaly or inflammatory reactions of the host [16]. The principal complications are hydatid cyst infection, biliary duct fistula, and rupture into the peritoneum or chest [17].

Ultrasounds are considered the best and the most convenient imaging modality for liver hydatid cysts [8]. They are a screening modality with a monitoring efficacy of treatment of up to 90% diagnostic accuracy, depending on the operator’s experience. An ultrasound can clearly demonstrate the hydatid sand, membranes, daughter cysts, and vesicles inside the hydatid cyst [18]. It can differentiate hydatid cyst Type I of the liver from simple liver cysts with 96% and 98% sensitivity and specificity, respectively. CTs can differentiate hydatid cyst Type 1 with 80% and 62% sensitivity and specificity, respectively [19].

A CT has a sensitivity rate of 94% to approach for liver hydatid [18] and is important in detecting calcification in the cyst wall or septa, demonstrating cystic structures, assessing complications, and in cases of obesity, excessive bowel gases, abdominal wall deformities, and previous surgery [13,18]. An MRI is another proper imaging modality for studying cystic components and the involvement of the biliary tree. The typical appearance of a cystic part is a low signal intensity on T1Wis, a markedly high signal on T2Wis, and a low signal intensity rim on T2WIs, which represents the collagen-rich pericyst layer, and is a characteristic sign of hydatid cysts [20]. An MRI is superior for demonstrating cyst wall defects, biliary communication, and differentiating hydatid cysts from simple cysts using diffusion-weighted sequences [20,21].

Hydatid cysts of the liver should be considered in the differential diagnosis, as they can mimic many cystic and even solid lesions of the liver, such as simple liver cysts, choledochal cysts, Caroli’s disease, hemangioendotheliomas, mesenchymal hamartomas, and teratomas [22].

The management of hydatid cysts includes medical therapy, percutaneous therapy, and surgical intervention. Medical therapy alone using mebendazole or albendazole has < 30% success rate [23]. Surgery remains the most effective therapy, with about 30% postoperative complications, such as biliocutaneous fistula and infection, and a demand for prolonged postoperative hospitalization [17]. The size of hydatid cysts is considered a significant predictor of morbidity and mortality, and the residual cavity is a challenging postsurgical problem in large hydatid cysts and is associated with a high risk of infection [23]. Minimally invasive methods have fewer complications and shorter hospitalizations [15]. Figure 3, Figure 4, Figure 5, Figure 6, Figure 7, Figure 8, Figure 9, Figure 10, Figure 11, Figure 13, Figure 14 and Figure 19.

### 5.2. Hydatid Cysts of the Lung

Hydatid cysts affect the human lung in 15% of cases [24] where the ingested eggs release oncosphere in the small bowel, which penetrates the bowel mucosa and enters the blood or lymph to reach the liver, lung, or other organs where they mature into hydatid cysts. The lower lobe is the most affected site of the lung, with a predilection for the posterior segments and the right lung. Pulmonary hydatid cysts are usually solitary and mostly unilateral. Multiple pulmonary hydatid cysts occur in 30% of cases, and bilateral cysts occur in 20% of cases [25].

A chest X-ray is the initial imaging modality; however, CTs and MRIs are considered the most useful modalities for imaging pulmonary hydatid cysts and their complications. Pulmonary hydatid cysts appear sharply defined, with smooth walls, round to oval shapes, homogeneous opacity, and in variable sizes [26]. Ruptured hydatid cysts with floating parasitic membranes in the fluid form the “water lily sign” or “camalote sign” [26,27]. Pulmonary hydatid cysts can progress asymptomatically for a long period, and perforation to the pleura can lead to life-threatening emergency cases, such as tension pneumothorax [28].

Intact lung hydatid cysts can be difficult to differentiate from other lung cysts. Lung hydatid cysts with crescent signs can mimic mycetomas, blood clots, Rasmussen aneurysms, and even lung carcinomas [29]. Surgery is the treatment of choice for pulmonary hydatid cysts and it can be safely performed without morbidity and negligible mortality rates, regardless of the hydatid size [30]. Figure 20 and Figure 21.

### 5.3. Hydatid Cysts of the Spleen

The spleen is the third most common organ affected by hydatid cysts, after the liver and lungs. As affection of the spleen is rare, it may create diagnostic challenges for physicians, especially when presenting as a simple cyst without classic imaging features. It can lead to disastrous complications, such as anaphylactic shock [31]. A CT is the most sensitive imaging modality for the diagnosis of splenic hydatid cysts and determining their number, size, and location [31,32]. The main diagnostic problem is that it can mimic other splenic cystic lesions, such as epidermoid and dermoid cysts, solitary abscesses, hematomas, cystic hemangiomas, intrasplenic pancreatic pseudocysts, and lymphangiomas [33]. A total splenectomy is the treatment of choice for splenic hydatid cysts [31,34]. Figure 10 and Figure 22.

### 5.4. Hydatid Cysts of the Brain

Hydatid cysts affect the human brain in 1–2% of cases [35]. Eighty percent of brain hydatids are observed in children, which is attributed to patent ductus arteriosus or valve dysfunctions, and typically affects the middle cerebral artery territory. Brain hydatids grow by 1–10 cm per year and they present with signs of increased intracranial pressure or focal neurological symptoms [36]. The clinical presentation of brain hydatid cysts depends on the size and location of the cysts. They commonly present with headaches and vomiting, hemiparesis, seizures, behavioral alterations, or even skull deformity [35,37]. Multiple brain hydatid cysts occur in more than 50% of patients and they are generally located in the cerebral hemispheres; however, they have been reported in the ventricles, epidural space, posterior fossa, pons, sella turcica, and parasellar region, and even in cavernous sinuses [38]. On medical imaging modalities, brain hydatids appear as well-circumscribed cystic lesions, rarely with daughter cysts. They present with CSF density on CT or signal intensity on an MRI, and usually with no surrounding edema or wall calcification. Hypointense rims on T2Wis are characteristic [39]. On T2Wis of MRIs, the presence of a low signal intensity rim around the cyst is characteristic [40]. Brain hydatid cysts can mimic cystic lesions, such as arachnoid cysts, porencephalic cysts, pyogenic abscesses, neurocysticercosis, and cystic tumors of the brain, and should be a differential diagnosis [39,41]. They can mimic cystic gliomas and difficult to differentiate from granulomas [42]. Figure 23.

### 5.5. Hydatid Cysts of the Musculoskeletal System

Hydatid cysts can be found anywhere in the human body, and they may show different imaging features according to their stage and complications [43]. They rarely involve bones and muscles; however, they can occur in a similar manner to many bone lesions, and they usually create a diagnostic issue preoperatively. An MRI is the preferred imaging modality for musculoskeletal hydatid cysts [44], which can resemble tumors and tumor-like lesions such as tuberculosis and fungal infections of the bones and muscles. A differential diagnosis depends on the patient’s history, medical imaging findings, and Echinococcus antibodies. A prompt diagnosis is of great importance in minimizing bone destruction and complications [45]. The diagnosis of hydatid cysts may be delayed till surgery and post-surgical biopsy. (Figure 24 and Figure 25, Table 1).

### 5.6. Rare Sites of Hydatid Cysts

Hydatid cysts can affect many organs in the body [46], including the thyroid gland, gallbladder, pancreas, uterus and adnexa, seminal vesicles, bones, muscles, skin, and subcutaneous tissue [47]. Hydatid cysts have been reported in the orbit as a retrobulbar cystic lesion that is confirmed by histopathology examination [48] in the left lobe of the thyroid gland mimicking colloid cysts and leading to left lobectomy and isthmectomy [49], and in the submandibular salivary gland mimicking enlarged lymph nodes diagnosed by fine-needle aspiration cytology [50]. They have been reported in parotid salivary glands leading to cystectomy with partial parotidectomy [51]; in the female breast, mimicking a breast mass forming diagnostic dilemma [52]; and in the male breast, mimicking a breast mass leading to lumpectomy where the diagnosis was confirmed on a histopathology examination [53]. They have been reported in the mediastinum mimicking mediastinal cystic lesions [54], and in the right ventricle of the heart mimicking intraventricular congenital cardiac cysts [55]. They have been reported in the pancreas presented by acute pancreatitis [56], in the adrenal gland presented with arterial hypertension and left flank pain, diagnosed on CT, and confirmed on surgery [57]. They have been reported in the uterus as a cystic mass leading to subtotal hysterectomy and confirmed by microscopic examination [58], in the ovary mimicking a pelvic mass [59], in the posterior triangle of the neck presented with a slow-growing painless mass for three years and diagnosed by fine-needle aspiration cytology [60]. Hydatid cysts also have been reported over the plantar surface of the foot and diagnosed intraoperatively by the typical appearance of a hydatid cyst [61], and even in the subcutaneous tissue of the face of a child, where the CT clinched the diagnosis [62].

## 6. Limitations

This review is limited in that the ultrasound images of the CE4 and CE5 stages were not available. An image of the larval stage of *E. granulosus* under a microscope was also not available.

## 7. Conclusions

Hydatid disease is a parasitic infection that predominantly affects the liver and followed by the lung, but it can affect almost any organ in the human body. According to the phase of growth, a hydatid cyst can occur in different sizes and shapes, which may mimic benign and even malignant neoplasms and may create diagnostic challenges in some cases. Hydatid cysts can mimic simple cysts, choledochal cysts, Caroli’s disease, mesenchymal hamartomas, or even metastasis of the liver. They can mimic lung cystic lesions, mycetomas, blood clots, Rasmussen aneurysms, and even lung carcinomas. Differential diagnosis can be difficult for arachnoid cysts, porencephalic cysts, pyogenic abscesses, and even cystic tumors of the brain, such as astrocytomas. They can create diagnostic dilemmas in the musculoskeletal system.

## Figures and Tables

**Figure 1 diagnostics-13-01127-f001:**
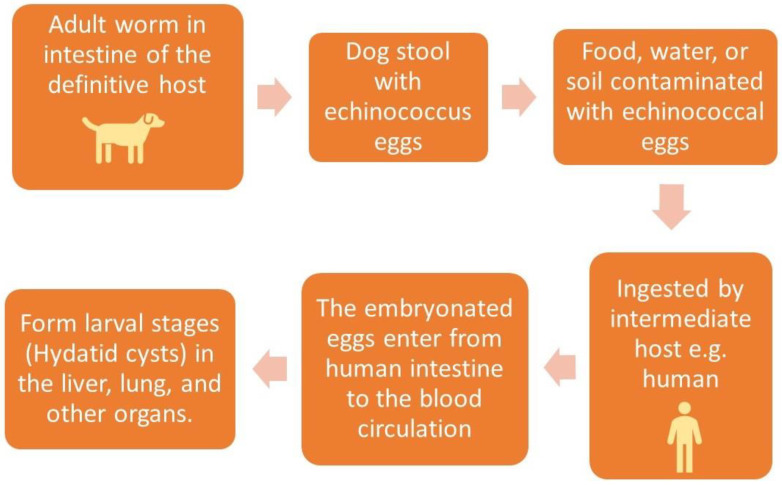
Method of human infection with *Echinococcus granulosus*.

**Figure 2 diagnostics-13-01127-f002:**
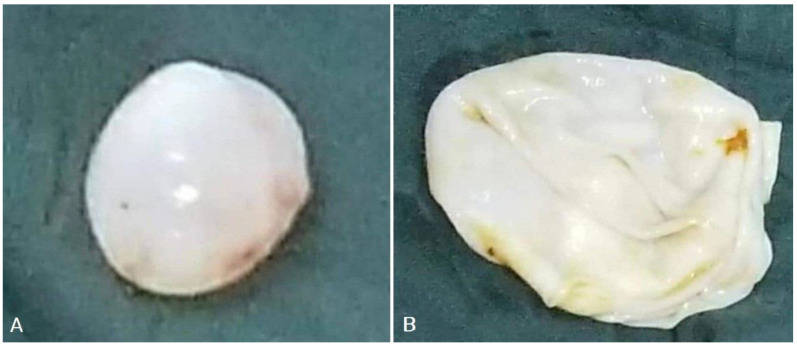
Images of gross hydatid cysts from outside (**A**), and inside (**B**) of two different cysts.

**Figure 3 diagnostics-13-01127-f003:**
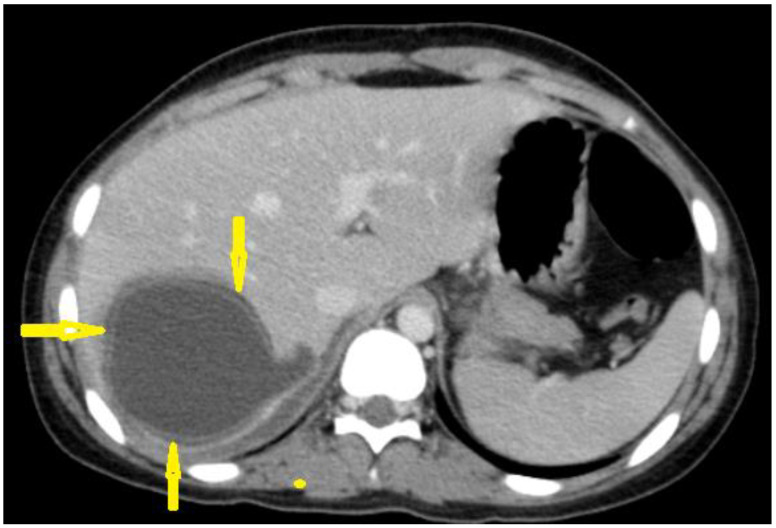
Contrast-enhanced computed tomography of a 15-year-old female presenting with abdominal pain showing a well-circumscribed, unilocular, non-enhanced cystic lesion in the right lobe of the liver with typical “double-wall sign” of hydatid cyst (arrows).

**Figure 4 diagnostics-13-01127-f004:**
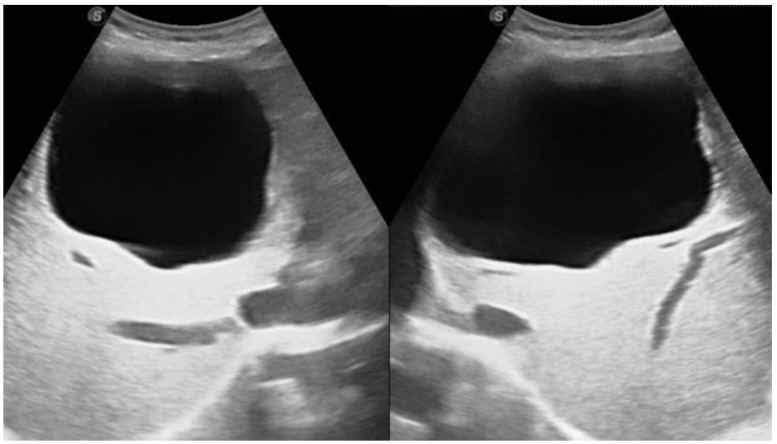
Ultrasound images of a 35-year-old female showing a well-circumscribed, unilocular, anechoic cystic lesion measuring 10 cm with no internal septation or solid component centered in segment IV of the right hepatic lobe, exerting slight compression on the gallbladder with no remarkable biliary dilatation or frank calcifications. Picture is in keeping with hydatid cyst stage CE1 (WHO classification).

**Figure 5 diagnostics-13-01127-f005:**
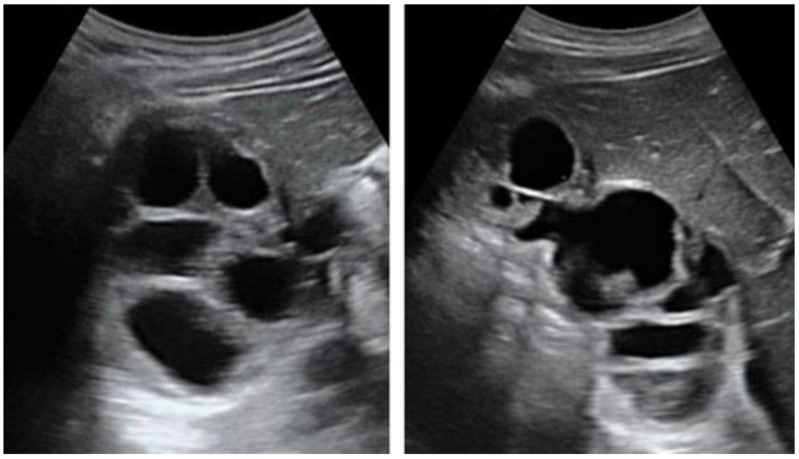
Ultrasound image of a 17-year-old female presenting with upper abdominal pain shows an 11 × 6 cm well-demarcated, mild lobulated margin with multiple internal septations of higher attenuation seen in segment VI of the right lower hepatic lobe, descended in front, lateral to the right kidney, and exerting mild kidney indentation. Picture is in keeping with the hydatid cyst CE2 (WHO classification).

**Figure 6 diagnostics-13-01127-f006:**
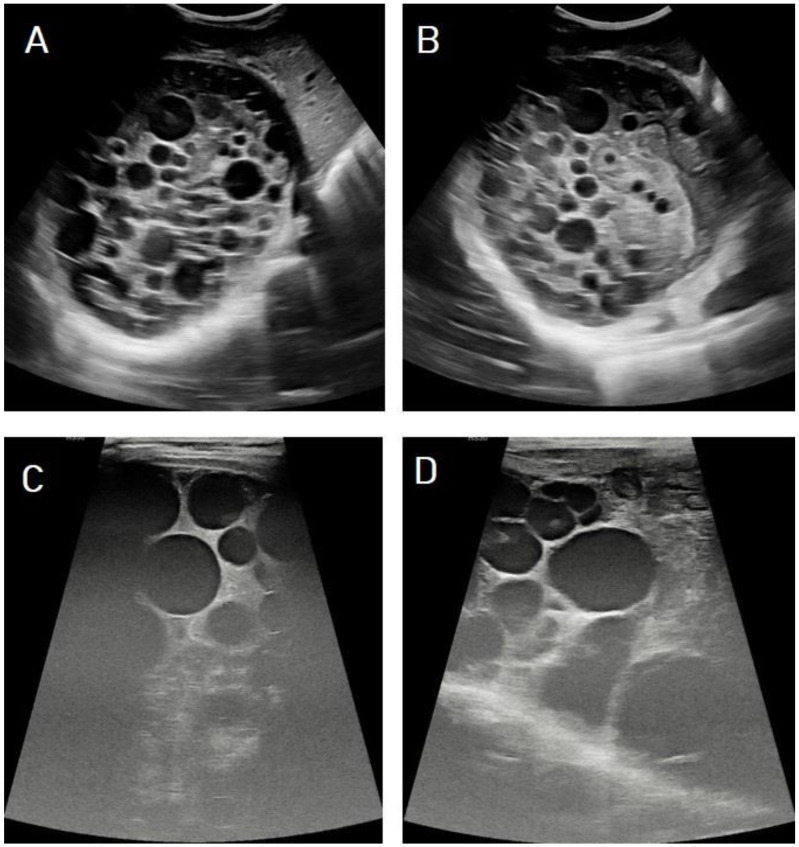
Ultrasound images of the liver of a 10-year-old male presenting with abdominal pain showing 15 × 12 cm well-circumscribed, cystic lesion thin echogenic wall replacing the V, VI, and VII segments of right hepatic lobe containing multiple daughter cysts and echogenic matrices (**A**,**B**), which are typical images of hydatid cysts in stage CE3b (WHO classification). The daughter cysts appeared more obviously on the high-resolution images of the high-frequency linear transducer (**C**,**D**).

**Figure 7 diagnostics-13-01127-f007:**
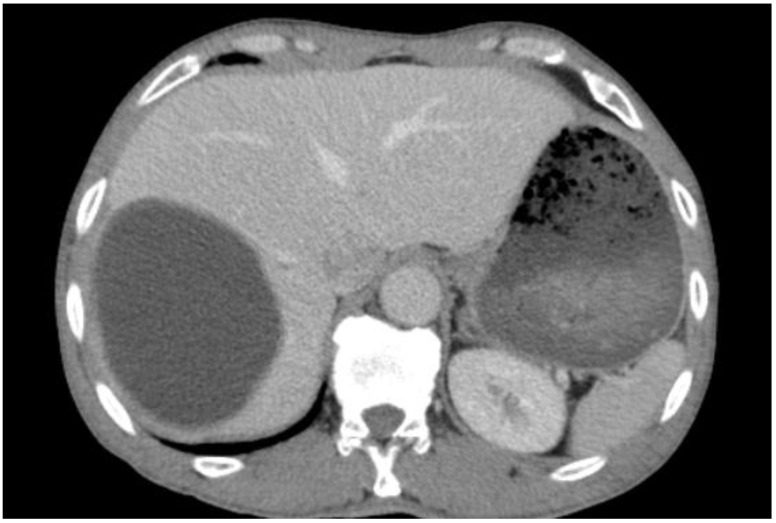
Selected CT images of a 35-year-old female showing a well-circumscribed, round, unilocular hypodense cystic lesion centered in segment IV of the right hepatic lobe, measuring 10 × 7.7 × 6.3 cm with no internal septation or solid component, no frank calcifications, or enhanced internal component after contrast administration, which is a typical image of hydatid cyst Type I.

**Figure 8 diagnostics-13-01127-f008:**
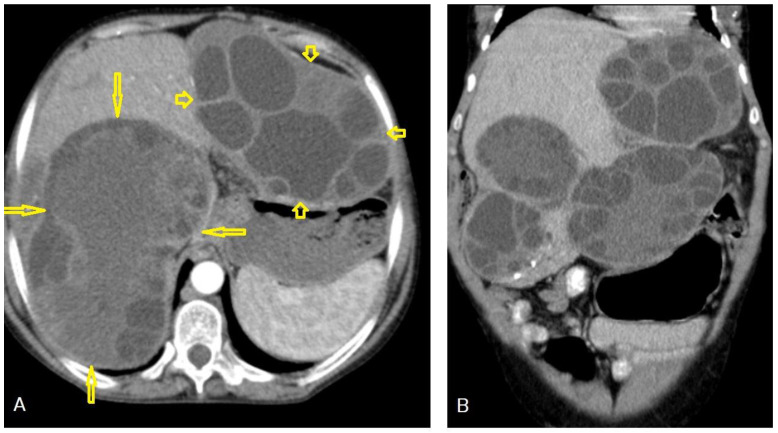
Selected axial (**A**) and coronal (**B**) CT images of a 60-year-old female showing multiple well-defined cystic lesions occupying the liver: about five lesions in the right lobe, where the largest one measured 17 × 9.2 × 15.3 cm (long arrows), occupying the right posterior lobe, and two cysts occupying the left lobe, where the largest one measured 11.4 × 10.3 × 8.7 cm (short arrows). Most of the lesions contain multiple vesicular cysts inside them (daughter cysts). The lesions exert a compression effect on the portal vein, stomach, and ascending colon, intending to reach the right kidney upper pole. This is a typical picture of an active-stage hydatid cyst Type IIA.

**Figure 9 diagnostics-13-01127-f009:**
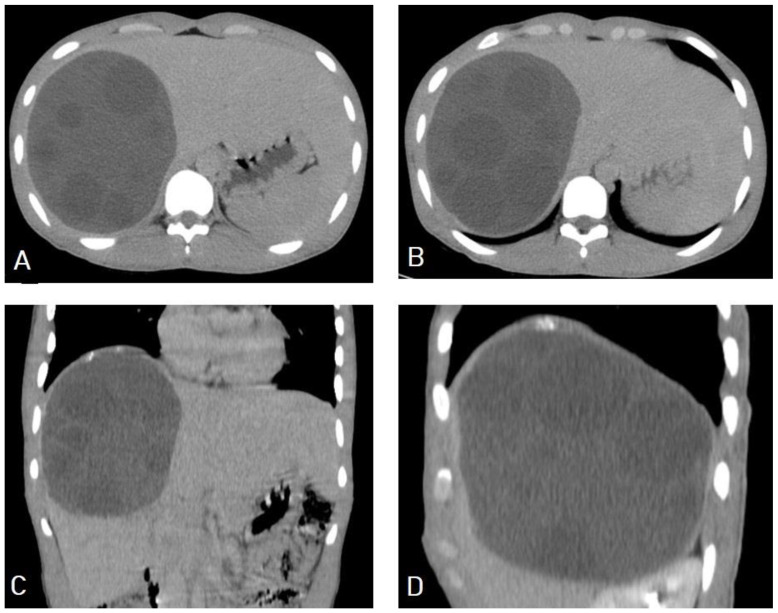
Selected axial (**A**,**B**) and coronal (**C**,**D**) CT images of a 16-year-old male presenting with abdominal pain and distension showing a well-defined cystic lesion in the right lobe of the liver, containing multiple vesicular cysts (daughter cysts) and hyperdense matrix, and forming the “spoke-wheel” appearance of a hydatid cyst.

**Figure 10 diagnostics-13-01127-f010:**
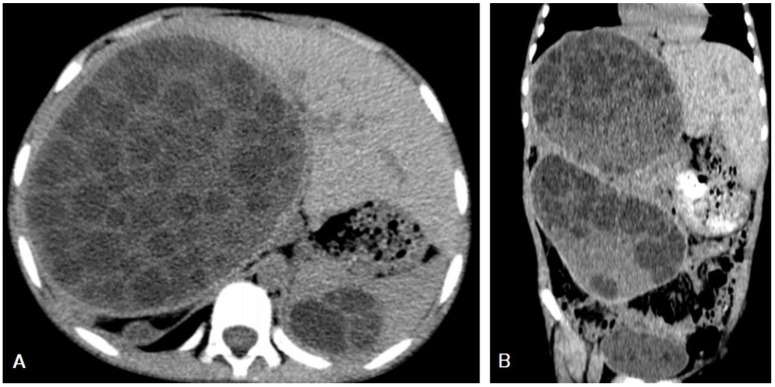
Selected axial (**A**) and sagittal (**B**) CT image of a 10-year-old male presenting with abdominal pain shows an enlarged liver with a 15 × 12 cm thin-wall cystic lesion seen replacing the V, VI, and VII segments of right hepatic lobe containing multiple daughter cysts, which is typical picture of a hydatid cyst with normal biliary tree and portal vein. Another similar thin-wall cystic lesion measuring 5.5 × 3.7 cm contains multiple daughter cysts, which is a typical picture of hydatid cysts of the spleen. Additionally, showing multiple similar cystic lesions measuring 11 × 16 cm in the right upper abdominal cavity causing mass effect in the displacement of the right kidney, and another three smaller cystic lesions in the right subphrenic space measuring 3.8 × 5.2 cm, 4 × 2 cm, and 4 × 1.5 cm. Moreover, showing another similar cystic lesion at the recto-vesical pouch measuring about 6 × 4 cm. The above findings represent a typical case of liver, spleen, and peritoneal hydatidosis and Type IIB.

**Figure 11 diagnostics-13-01127-f011:**
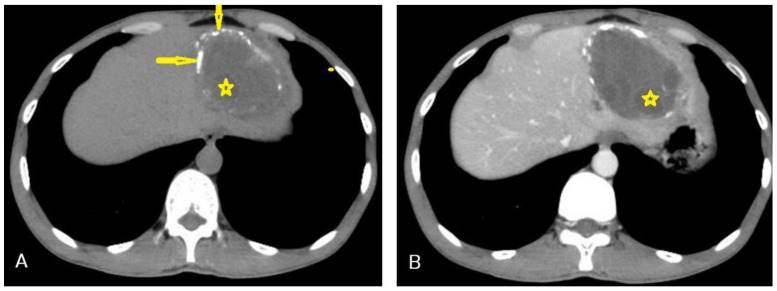
Selected axial non-enhanced (**A**), and contrast-enhanced (**B**) CT images of the abdomen of an adult male patient showing a well-circumscribed hypodense cystic lesion with multiple daughter cysts inside it (stars), with stippled calcification of the wall (arrow), and with no enhancement after contrast administration. Picture of hydatid cyst Type IIC.

**Figure 12 diagnostics-13-01127-f012:**
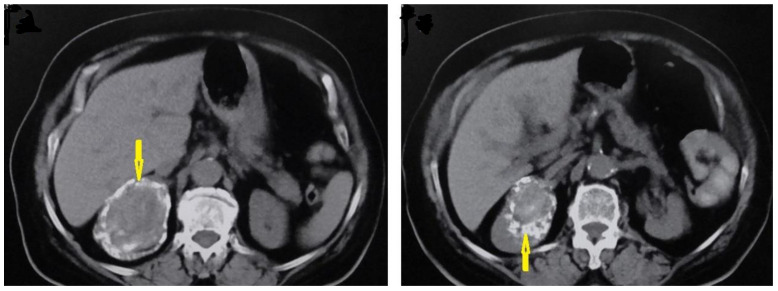
Selected axial images of abdominal CT of a 60-year-old female showing a well-circumscribed unilocular cystic lesion in the right kidney with calcified margins (arrows), no mass effect, and no daughter cysts inside it, which is typical of hydatid cyst Type III.

**Figure 13 diagnostics-13-01127-f013:**
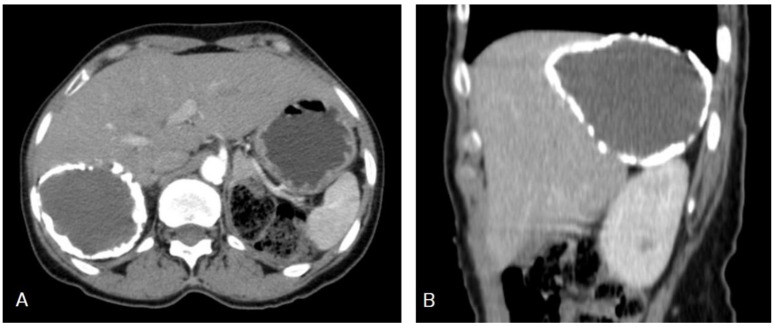
Selected axial (**A**) and sagittal (**B**) CT images of a 50-year-old female presenting with right upper quadrant pain, showing a well-circumscribed unilocular cystic lesion in the right lobe of the liver with calcified margins and no daughter cysts inside it, which is typical of inactive hydatid cyst Type III.

**Figure 14 diagnostics-13-01127-f014:**
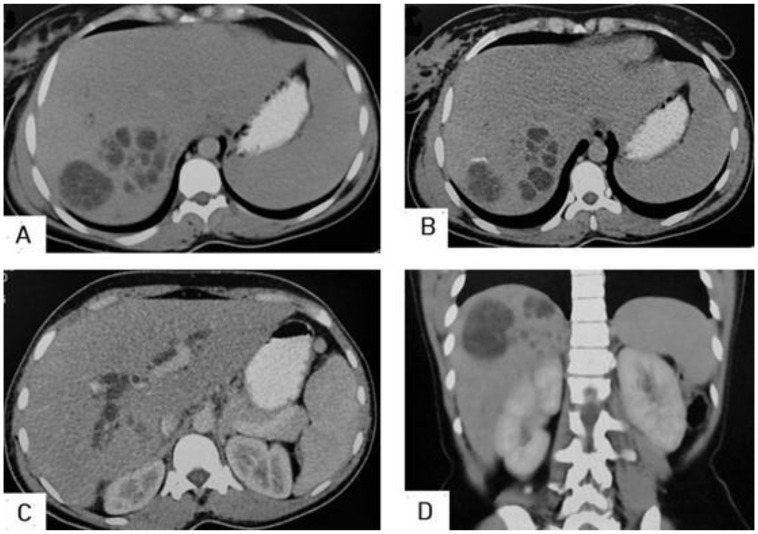
Selected CT images of a 26-year-old male showing multiple well-defined grouped cysts in the right lobe of the liver, the largest about 4 cm, seen on axial (**A**) section, with partial peripheral calcification seen on axial (**B**) section. There are multiple irregular, ill-defined small cysts around the confluence of the main hepatic bile ducts seen on axial (**C**) section with moderate dilatation of the common bile duct and mild dilatation of the intrahepatic biliary tree seen on coronal (**D**) section. Picture is in keeping with biliary tree seeding from the liver hydatid cyst.

**Figure 15 diagnostics-13-01127-f015:**
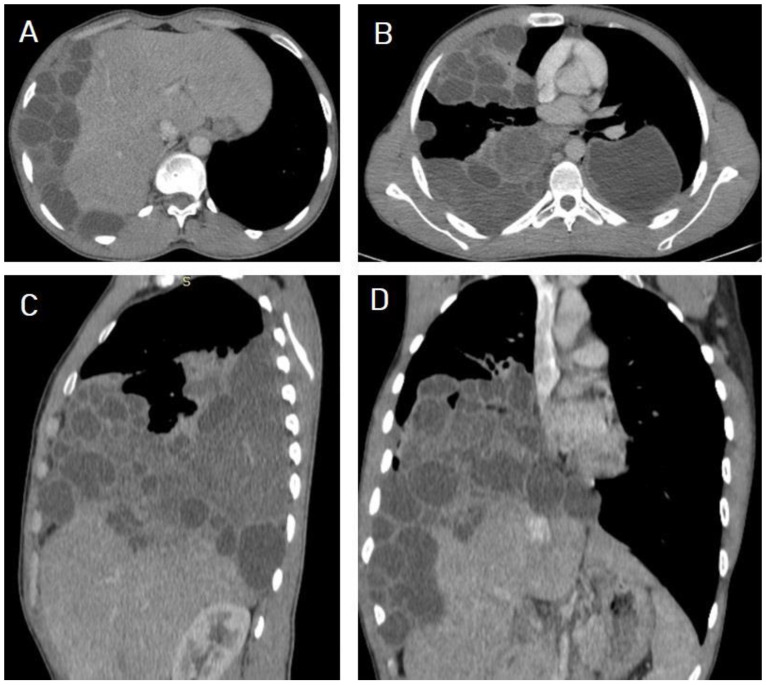
Selected CT images of a 40-year-old male presenting with cough and dyspnea showing a ruptured liver hydatid cyst complicated with disastrous dissemination into the peritoneal cavity seen on the axial section (**A**), and bilateral chest seeding seen on axial (**B**), sagittal (**C**), and coronal (**D**) sections. Picture is in keeping with peritoneal and bilateral chest seeding from the liver hydatid cyst.

**Figure 16 diagnostics-13-01127-f016:**
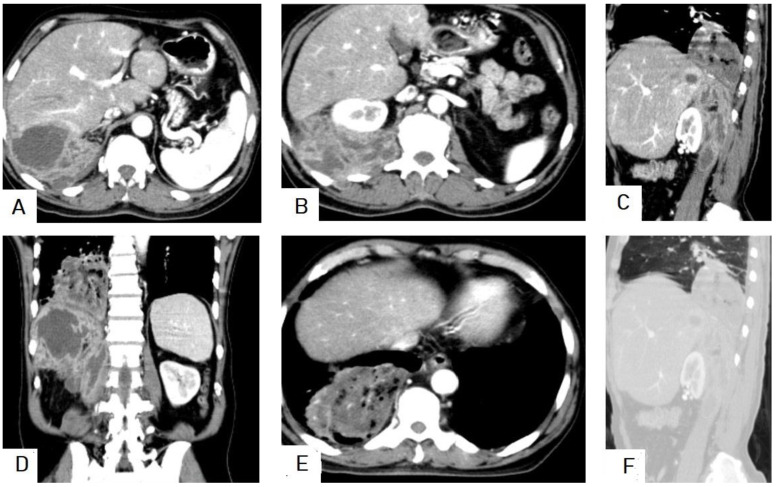
Selected CT images of an 80-year-old man presenting with chest pain, showing about 6.8 × 4 cm multiloculated and peripherally enhanced collection and seeming to arise from the subcapsular segment-VII of the right lobe of liver with disrupted capsule and exophytic tracking component to subphrenic region into the posterior perinephric and pararenal spaces seen on axial (**A**,**B**) sections, and extending across the proximal psoas muscle for 7.5 cm caudally seen on sagittal (**C**) and coronal (**D**) sections. Superiorly, the lesion tracks the right hemidiaphragm into the lung, where there is a well-circumscribed mass-like lesion measured at 8.7 × 6.3 × 5 cm with thickened wall with obvious bronchovascular markings seen extended within the lesion in the adjacent lower lobe area seen on sagittal (**C**), coronal (**D**), axial (**E**), and sagittal lung window (**F**) sections. The picture suggests ruptured hepatic hydatid cyst tracking to the retroperitoneaum, psoas muscle, and lung, with superimposed psoas abscess formation.

**Figure 17 diagnostics-13-01127-f017:**
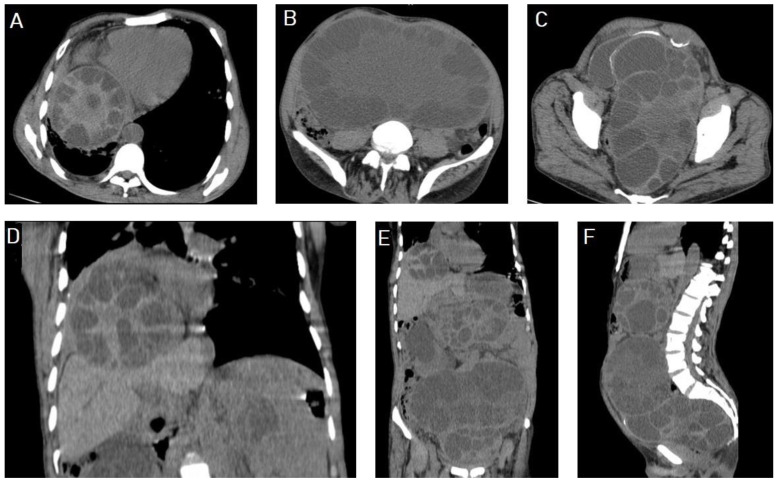
Selected CT images of a 60-year-old male with renal impairment presenting with abdominal pain and distention, showing 8.6 × 8.2 cm hydatid cyst with multiple daughter cysts in the lower lobe of the right lung, indenting the right hemidiaphragm, and compressing the lower liver seen on axial (**A**) and coronal (**D**) sections. Additionally, showing multiple abdominal and pelvic large hydatid cysts with multiple internal daughter cysts compressing bilateral ureters, causing moderate bilateral hydronephrosis, and compressing the urinary bladder anteriorly seen on axial (**B**,**C**), coronal (**D**,**E**), and sagittal (**F**) sections. Pictures from Type IIA with typical “spoke-wheel appearance” occurring on images (**A**–**D**).

**Figure 18 diagnostics-13-01127-f018:**
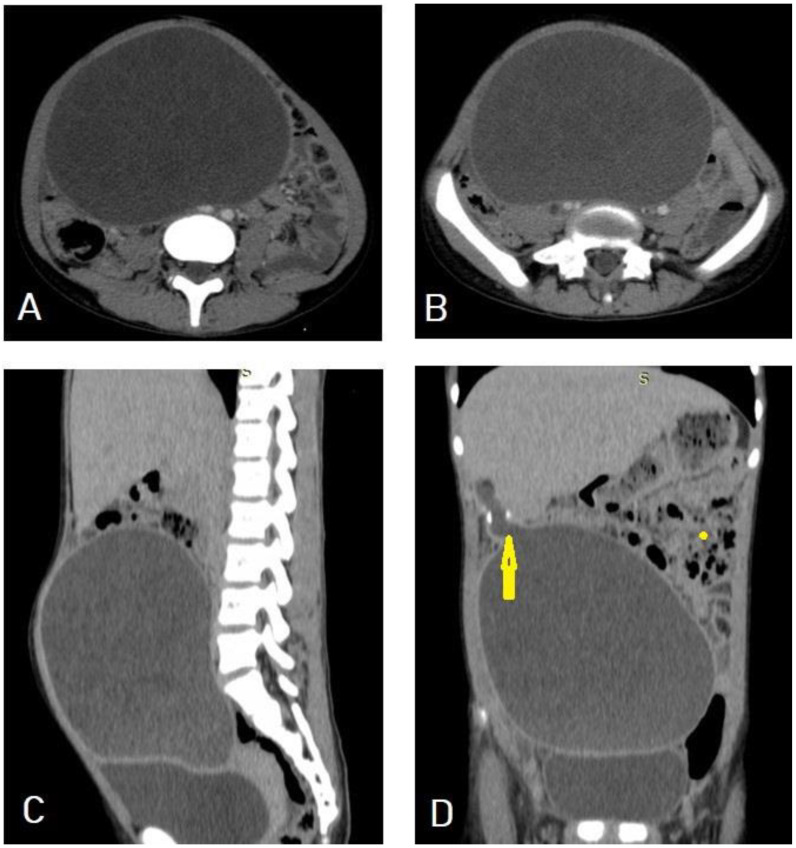
Selected CT images of a 13-year-old patient presenting with abdominal distention and feeling mass showing a 16 × 15 × 11 cm well-circumscribed, multiloculated cystic lesion in the right side of abdominal and upper pelvic cavities as seen in axial (**A**,**B**) sections, exerting mass effect in the form of bilateral mild hydronephrosis, displacing the bowel loops laterally, and compressing the urinary bladder as seen in the sagittal (**C**) and coronal (**D**) sections. The cyst had a narrow neck-like (arrow) at the superolateral aspect connected with other cystic lesions in segment-VI of the liver. The above findings are in keeping with a large abdominal hydatid cyst.

**Figure 19 diagnostics-13-01127-f019:**
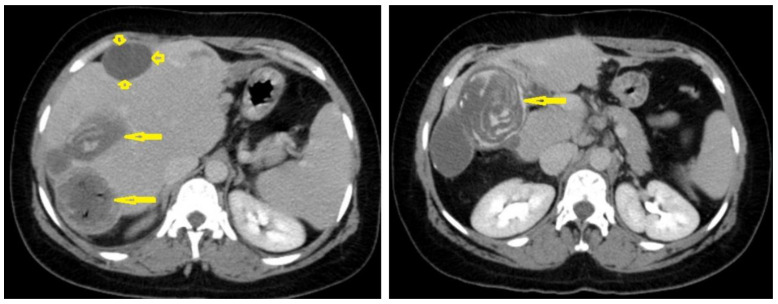
Selected axial sections of CT images of a 30-year-old female with a previous surgery for a liver hydatid cyst presenting with abdominal pain showing remnants of the hydatid cyst (arrows) and another hydatid cyst (arrow heads).

**Figure 20 diagnostics-13-01127-f020:**
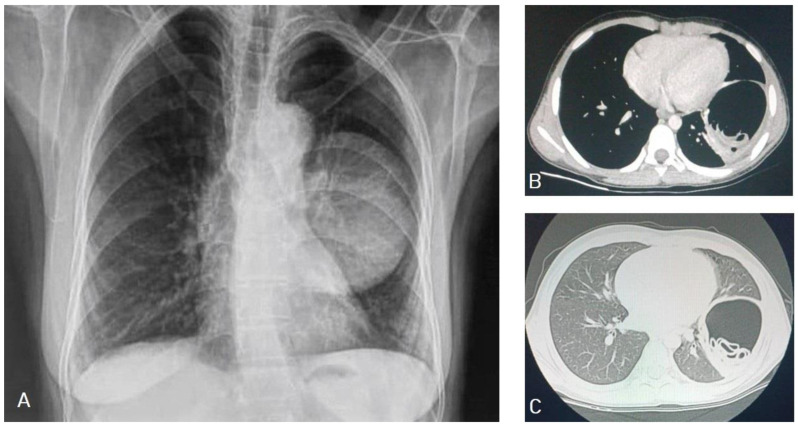
Chest X-ray of a 9-year-old male patient showing a well-defined oval shape with smooth margins and a large mass in the left lung (**A**). Selected axial CT images, mediastinal window (**B**), and lung window (**C**) show a well-circumscribed smooth margin intrapulmonary mass measuring 8.4 × 6.4 cm with fluid and gas contents with twisted-linear structures floating within the fluid contents of the cyst, forming a “water lily sign”, and representing hydatid cyst with detachment of the germinal membrane of the endocyst. The mass causes a mass effect in the form of compression of the lower lobe of the left lung.

**Figure 21 diagnostics-13-01127-f021:**
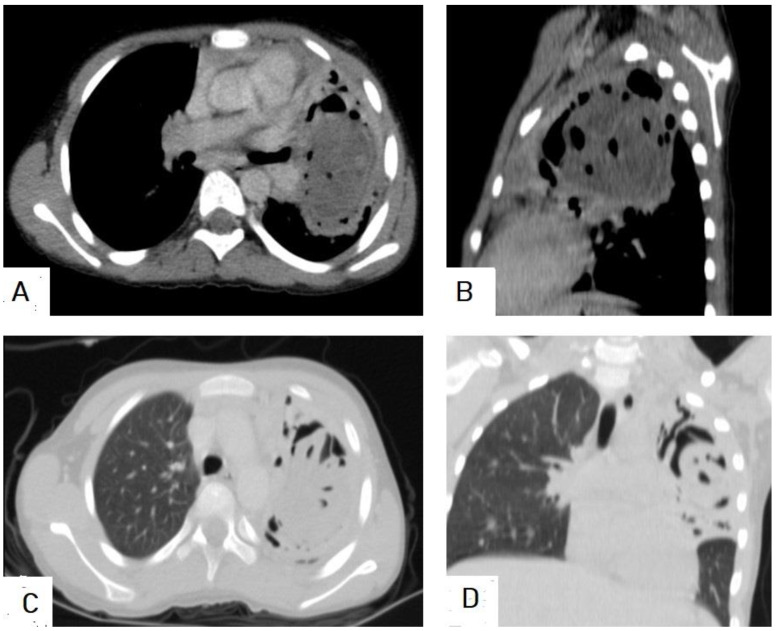
Selected sections of chest CT images of a 10-year-old female presenting with cough, dyspnea, and hemoptysis showing a hypodense lesion in the left lung with thick-enhancing wall seen on axial (**A**) and sagittal (**B**) mediastinal window sections. The “air-bubble” sign inside the lesion is seen in the mediastinal window sections (**A**,**B**), and on the lung window axial (**C**) and coronal (**D**) sections. Picture is in keeping with ruptured hydatid cyst.

**Figure 22 diagnostics-13-01127-f022:**
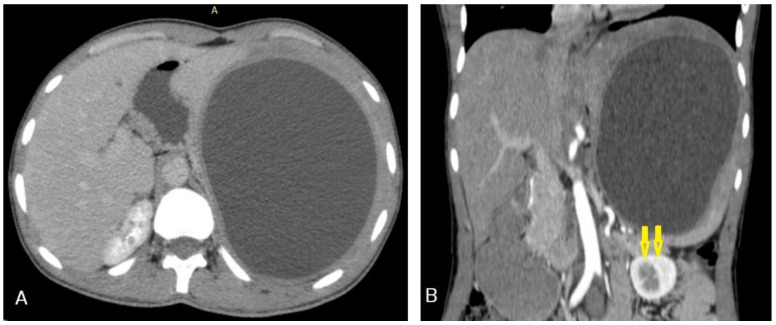
Selected axial (**A**) and coronal (**B**) CT images of a 25-year-old male with left upper abdominal pain and distention, showing a 15 × 13 cm well-circumscribed, unilocular cystic lesion in the spleen that causes a mass effect in the form of compression and downward displacement of the left kidney (arrows). The lesion showed homogeneous contents, no septations, no calcification, and no enhancement after contrast administration, which is a typical picture of a hydatid cyst of the spleen.

**Figure 23 diagnostics-13-01127-f023:**
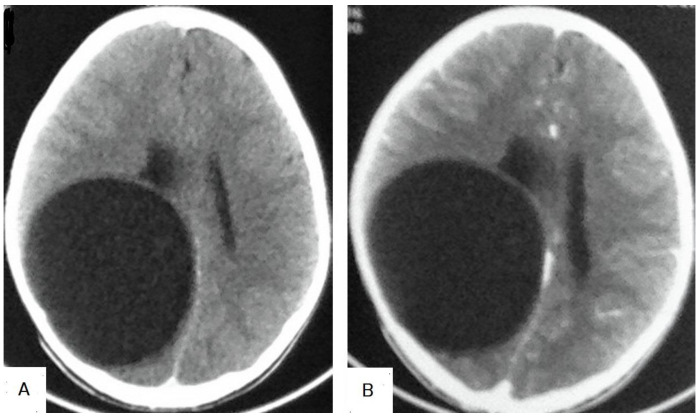
Selected CT images of a 12-year-old male presenting with signs of increased intracranial pressure, showing a large well-circumscribed cyst in the right cerebral hemisphere that caused a mass effect in the form of compression of the ipsilateral lateral ventricle complicating with ipsilateral hydrocephalus. The cyst showed no surrounding edema, no calcification (**A**), and no enhancement after contrast administration (**B**), which is a typical feature of cerebral hydatid cysts.

**Figure 24 diagnostics-13-01127-f024:**
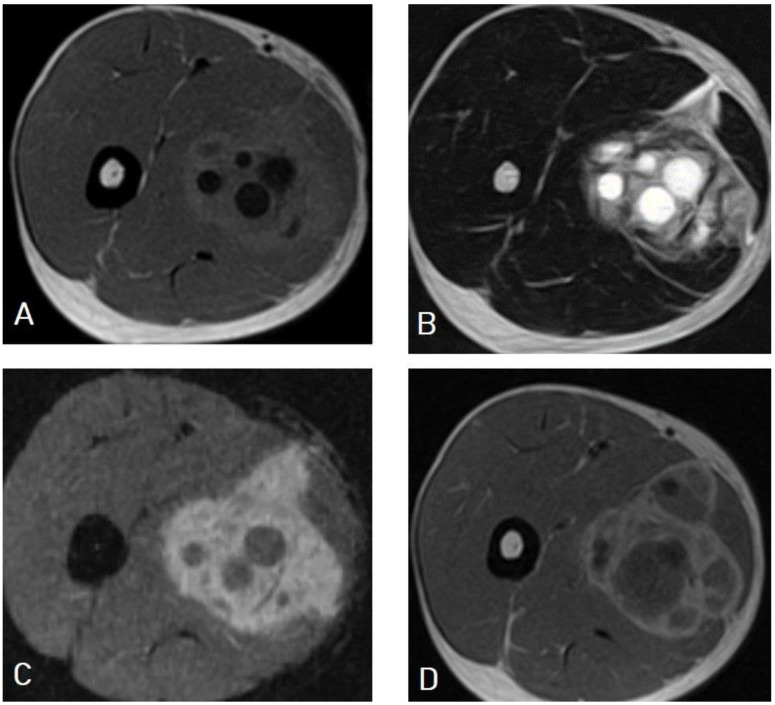
Selected images of thigh MRI of a 23-year-old male showing an 11 × 8 × 8 cm heterogeneous complex cystic lesion within the anteromedial intramuscular compartment of the upper right thigh with thickened edematous adjacent subcutaneous fat containing multiple small daughter cysts appearing hypointense on T1-weighted images (**A**), hyperintense on T2-weighted images (**B**), hyperintense on fat-suppression sequences (**C**), and marginal wall enhancement on T1 with contrast (**D**), which is typical appearance of intramuscular hydatid cyst with superimposed infection and impending rapture.

**Figure 25 diagnostics-13-01127-f025:**
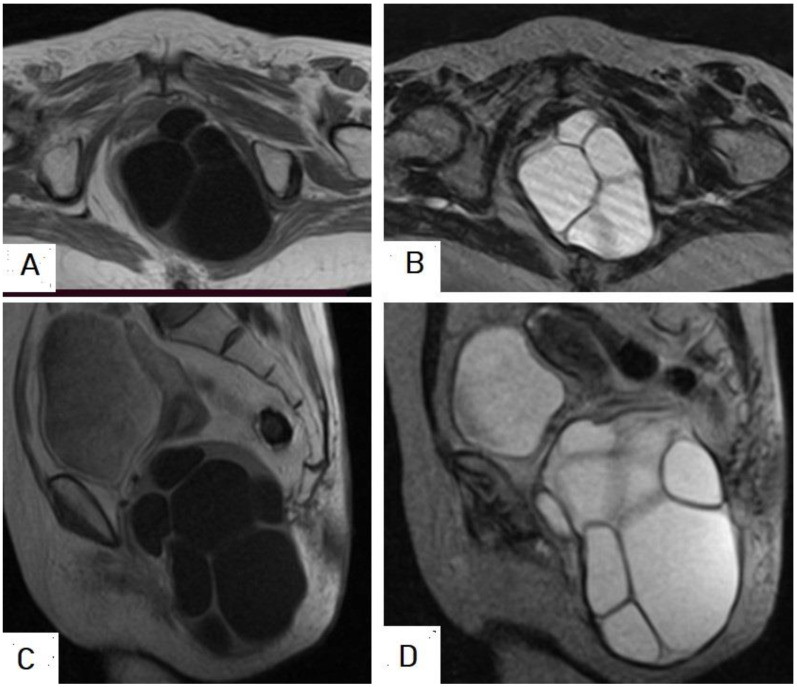
Selected images of pelvic MRI of a 75-year-old female presenting with pelvic pain, difficulty defecating, and micturition, which shows a 12 × 9 cm well-circumscribed, multiloculated cystic lesion in the left presacral and paranal, with ischiorectal fossa causing mass effect in the form of displacing the anal canal and rectum to the right side. The lesion appears hypointense on axial and sagittal T1-weighted images (**A**,**C**), and hyperintense on axial and sagittal T2-weighted images (**B**,**D**), with multiple cysts inside it, which is a typical picture of an intramuscular hydatid cyst.

**Table 1 diagnostics-13-01127-t001:** List of differential diagnoses of hydatid cysts.

Reference	Differential Diagnosis of the Hydatid Cysts
Erdem et al. [22]	Liver hydatid cyst can mimic multiple cystic or solid lesions of the liver, such as the following: 1: Simple liver cysts. 2: Choledochal cysts. 3: Caroli’s disease. 4: Hemangioendotheliomas. 5: Mesenchymal hamartomas. 6: Teratomas.
Garg et al. [29]	I: Intact lung hydatid cyst can mimic the following: 1: Any lung cysts. II: Lung hydatid cysts with crescent signs can mimic the following: 1: Mycetomas. 2: Blood clots. 3: Rasmussen aneurysms. 4: Lung carcinomas.
Merad et al. [33]	Hydatid cysts of the spleen can mimic other splenic cystic lesions, such as the following: 1: Epidermoid cysts. 2: Epidermoid cysts. 3: Solitary abscesses. 4: Hematomas. 5: Cystic hemangiomas. 6: Intrasplenic pancreatic pseudocysts. 7: Lymphangiomas.
Abbasi et al. [39] Agrawal et al. [41]	Brain hydatid cysts can mimic brain cystic lesions, such as the following: 1: Arachnoid cysts. 2: Porencephalic cysts. 3: Pyogenic abscesses. 4: Cystic tumors of the brain. 5-Neurocysticercosis.
Ganjeifar et al. [42]	Brain hydatid cysts can mimic brain lesions, such as the following: 1: Pyogenic abscesses. 2: Granulomas. 3: Cystic gliomas. 4: Epidermoid cysts. 5: Arachnoid cysts.
Togral et al. [45]	Hydatid cyst of the musculoskeletal system can resemble tumors and tumor-like lesions, such as the following: 1: Tuberculosis infections of the bones and muscles. 2: Fungal infections of the bones. 3: Others.

## Data Availability

Data available on request.

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
