# Peer review of "Hydatid Disease: A Radiological Pictorial Review of a Great Neoplasms Mimicker"

_diagnostics, 2023, doi:10.3390/diagnostics13061127_

Round 1

Reviewer 1 Report

Thanks for this comprehensive review on Hydatid Disease. It will add an important dimension to the literature on this important parasite. 

I have suggested some improvements in the file attached. 

Author Response

Reply is in the uploaded word file.

Reviewer 2 Report

 This is a well-written and exquisitely illustrated paper that should be of great value to imaging specialists throughout the world.  Echinococcosis remains a rare clinical presentation in most countries, yet episodic cases are to be expected as a consequence of international travel, etc.  This paper will serve as a textbook quality reference for those who encounter such cases only rarely. 

 This reviewer had only a few, very minor suggestions of a purely editorial nature. 

 Line 18.  E. granulosus is a species name, not a genus. 

 Lines 43 and 44.  Genus and species names, by scientific convention, are italicized when appearing in print.  However, the editors can determine whether the present variations are consistent with the editorial policies of Diagnostics. 

 Figure 1.  “echinococcus” in the second frame should be capitalized and italicized.

 Legend, Figure 18.  This paragraph of text seems to be presented in at least two different fonts. 

 Abbreviations:   Similar comments regarding italicization

Author Response

Reply is in the uploaded word file.
